# High throughput pMHC-I tetramer library production using chaperone-mediated peptide exchange

Sarah A. Overall[1,5], Jugmohit S. Toor[1,5], Stephanie Hao[2], Mark Yarmarkovich [3], Sara M. O'Rourke[1], Giora I. Morozov[1], Son Nguyen [4], Alberto Sada Japp [4], Nicolas Gonzalez[1], Danai Moschidi[1], Michael R. Betts[4], John M. Maris[3], Peter Smibert [2] & Nikolaos G. Sgourakis[1✉]

Peptide exchange technologies are essential for the generation of pMHC-multimer libraries used to probe diverse, polyclonal TCR repertoires in various settings. Here, using the molecular chaperone TAPBPR, we develop a robust method for the capture of stable, empty MHC-I molecules comprising murine H2 and human HLA alleles, which can be readily tetramerized and loaded with peptides of choice in a high-throughput manner. Alternatively, catalytic amounts of TAPBPR can be used to exchange placeholder peptides with high affinity peptides of interest. Using the same system, we describe high throughput assays to validate binding of multiple candidate peptides on empty MHC-I/TAPBPR complexes. Combined with tetramer-barcoding via a multi-modal cellular indexing technology, ECCITE-seq, our approach allows a combined analysis of TCR repertoires and other T cell transcription profiles together with their cognate antigen specificities in a single experiment. The new approach allows TCR/pMHC interactions to be interrogated easily at large scale.

[1] Department of Chemistry and Biochemistry, University of California Santa Cruz, 1156 High St., Santa Cruz, CA 95064, USA. [2] Technology Innovation Lab, New York Genome Center, 101 6th Ave, New York, NY 10013, USA. [3] Division of Oncology, Center for Childhood Cancer Research, Children's Hospital of Philadelphia and Department of Pediatrics, Perelman School of Medicine, University of Pennsylvania, 3400 Civic Center Blvd, Philadelphia, PA 19104, USA. [4] Department of Microbiology, Perelman School of Medicine, University of Pennsylvania, 3400 Civic Center Blvd, Philadelphia, PA 19104, USA. [5]These authors contributed equally: Sarah A. Overall, Jugmohit S. Toor. ✉email: nsgourak@ucsc.edu

T cells recognize foreign or aberrant antigens presented by MHC-I expressing cells through the T cell receptor (TCR) and is the first critical step towards establishment of protective immunity against viruses and tumors[1]. Fluorescently tagged, multivalent MHC class-I reagents (multimers) displaying individual peptides of interest have revolutionized detection of antigen specific T cells[2]. Staining with multimers followed by flow cytometry is routinely used to interrogate T cell responses, to characterize antigen-specific TCR repertoires and to identify immunodominant clones[3–5]. However, polyclonal repertoires are estimated to contain $10^5$–$10^8$ TCRs of distinct antigen specificities[6]. Preparing libraries of properly conformed peptide/MHC-I (pMHC-I) molecules displaying an array of epitopic peptides to probe such repertoires remains a significant challenge, due to the inherent instability of empty (i.e. peptide deficient) MHC-I molecules. To circumvent the problem of unstable peptide deficient MHC-I molecules, conditional MHC class I ligands are used[7]. Conditional ligands, bound to the MHC-I, can be cleaved by exposure to UV light[8], or to increased temperature[9]. Upon cleavage and in the presence of a peptide of interest, a net exchange occurs where the cleaved conditional ligand dissociates and the peptide of interest associates with the MHC-I, thereby forming the desired pMHC-I complex. Such conditional ligands, however, also have limitations. The use of photo-cleavable peptides necessitates a more elaborate protein purification protocol, and may lead to increased aggregation and sample loss during the peptide exchange step. Dipeptides, which compete with the C-terminus of bound peptides to promote exchange[10], and the stabilization of empty MHC-I molecules using an engineered disulfide bond[11] have been recently proposed as alternatives to catalyze peptide exchange under physiological conditions.

TAPasin Binding Protein Related (TAPBPR) is a chaperone protein homologue of Tapasin involved in the quality control of pMHC-I molecules[12]. TAPBPR associates with MHC-I molecules to edit the repertoire of displayed peptides at the cell surface[13]. In a similar manner to Tapasin[14], TAPBPR binds several MHC-I alleles in vitro to promote the exchange of low- for high-affinity peptides[15]. Using solution NMR, we recently performed a detailed characterization of the TAPBPR peptide exchange cycle for the murine H-2D$^d$ molecule[16]. This work revealed a critical role for N-terminal peptide interactions with the MHC-I peptide-binding groove, which allosterically regulates TAPBPR release from the pMHC-I. Peptide binding is therefore negatively coupled to chaperone release and, conversely, the affinity of incoming peptides for the MHC-I groove is decreased by 100-fold in the presence of TAPBPR[16], due to a widening of the MHC-I groove, as directly observed in X-ray structures of MHC-I/TAPBPR complexes[17,18]. In follow up work[19], we have elucidated the molecular determinants which confer specificity of interactions between TAPBPR and different class-I MHC alleles. This work revealed an approximately 5% minor conformation with a widened peptide-binding groove and altered dynamics at the α3/β2m interface exhibited by some MHC alleles, which enables binding to TAPBPR, consistently with reports that polymorphisms within the F-pocket of the groove relate to recognition by Tapasin[20] and TAPBPR[21].

Here, we leverage these mechanistic insights to design conditional ligands for the production of peptide deficient MHC-I/TAPBPR complexes for multiple murine and human MHC-I allotypes, independently of photo-cleavable peptides. Empty MHC-I/TAPBPR complexes are stable for months, can be readily multimerized and loaded with peptides of interest in a high-throughput manner. Focusing on a common human allele, HLA-A*02:01, we have extended the capability of our system by incorporating our multi-modal cellular indexing technology (ECCITE-seq)[22,23]. The resulting library of barcoded, TAPBPR-exchanged tetramers can be directly applied in a multiplexed analysis of numerous antigen-specificities simultaneously, enabling the identification of TCR V(D)J sequences together with other T cell transcriptional markers of interest in a single-cell format (Fig. 1). The use of a protein chaperone to accomplish peptide exchange, a task previously addressed using UV-sensitive synthetic peptide ligands[8], in vitro translation systems[24] or disulfide-linked MHC molecules[11], leads to clear gains in terms of ease of use, efficiency and sensitivity. Moreover, our indexing design allows a convenient workflow that is fully compatible with a commercially available kit, thereby allowing TCR specificities to be interrogated easily at large scale. Given the central role of T cell responses, our work is of immediate practical relevance to experimental immunologists and cancer biologists, with clear biomedical ramifications toward the development of T cell-based diagnostics and autologous therapies.

## Results

**Capturing empty MHC-I/TAPBPR complexes for murine H2 alleles.** To circumvent the need for photo-cleavable ligands, previously used to demonstrate high-affinity TAPBPR binding to empty MHC-I molecules[15], we explored the use of destabilizing placeholder peptides. We recently described a destabilizing N-terminally truncated mutant of the P18-I10 peptide_ GPGRAFVTI (gP18-I10, Fig. 2a)[16]. gP18-I10 showed a high affinity for a free H-2D$^d$ groove during in vitro refolding, but dissociated in the presence of TAPBPR to generate stable, empty H-2D$^d$/TAPBPR complexes (Supplementary Fig. 1a and d). In contrast, full-length P18-I10 remained captured in the groove of the H-2D$^d$/TAPBPR complex (Supplementary Fig. 1a–b). The previously reported 100-fold increase in the peptide off-rate for gP18-I10 relative to full-length P18-I10 peptide[16] resulted from the loss of specific polar contacts in the H-2D$^d$ A-pocket (Fig. 2a), as further reflected in predicted IC$_{50}$ values (24 μM gP18-I10 vs 23 nM for P18-I10). We have therefore termed this destabilizing peptide, a goldilocks peptide (gP18-I10). Extending the same concept to a different murine MHC-I molecule, H-2L$^d$, we tested an N-terminal truncation of the high-affinity p29 nonamer_ PNVNIHNF (IC$_{50}$ of 16.5 μM). However, the resulting 8mer peptide remained bound to the H-2L$^d$/TAPBPR complex (Supplementary Fig. 1a and e, Fig. 2b), indicating that truncation of the extreme N-terminal residue cannot be used as a general rule to generate goldilocks peptides (Supplementary Fig. 1e). As p29 fits the typical H-2L$^d$ binding motif of xPxx[NA]xx[FLM], we explored QLSPFPFDL (QL9), a predicted low-affinity peptide (IC$_{50}$ of 9.27 μM) with non-canonical Leu and Phe residues at position 2 and 5, respectively. Using QL9 as a placeholder peptide, we obtained empty H-2L$^d$/TAPBPR complexes by further adding 10 mM Gly-Phe dipeptide (GF), which promotes peptide release by directly competing for interactions in the F-pocket of the peptide-binding groove[10] (Supplementary Fig. 1f–g, Fig. 2b). Taken together, these results establish the principle that destabilization of peptide interactions at both ends of the groove, through a range of approaches, can be used to generate peptide-deficient MHC-I/TAPBPR complexes.

**TAPBPR-mediated peptide exchange on relevant HLA alleles.** To extend these findings using murine MHC-I alleles towards a high-throughput method for the production of pMHC tetramer libraries for human alleles, we focused on the common allele HLA-A*02:01 which displays a wide range of immunodominant viral and tumor epitopes, rendering the study of HLA-A*02:01-restricted responses highly relevant[25]. Guided by the existing TAX/HLA-A*02:01 crystal structures[26], we designed a number of

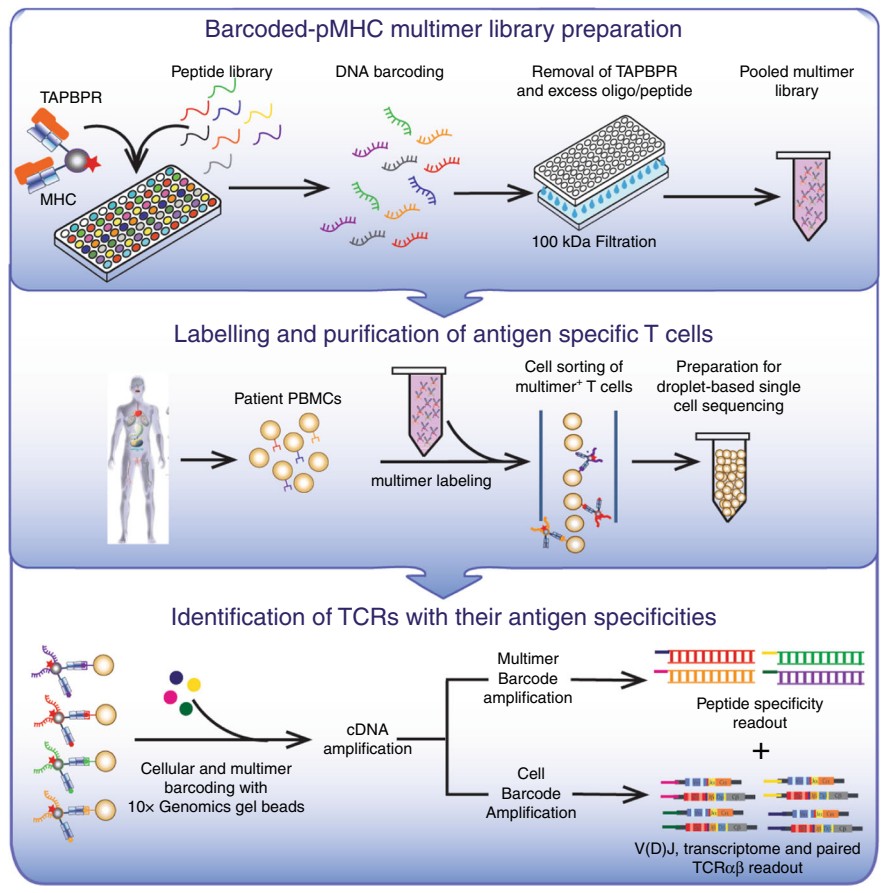

**Fig. 1 Linking peptide specificities with T cell transcriptomes.** Fluorophore-labeled, empty MHC-I/TAPBPR multimers are loaded with peptides of interest on a 96-well plate format and individually barcoded with DNA oligos designed for 10x Genomics and Illumina compatibility. TAPBPR and excess peptide, along with free oligo, are removed by centrifugation and the multimers are pooled together. A single patient sample can be stained with the pooled multimer library, and collected by fluorescence-activated cell sorting. Tetramer associated oligos and cellular mRNA from individual cells are then barcoded using 10x Genomics gel beads, followed by cDNA synthesis, library preparation and library sequencing. This workflow enables the transcriptome and paired αβ TCR sequences to be linked with pMHC specificities in a single experiment.

variants of the LLFGYPVYV (TAX) peptide by progressively reducing N-terminal polar contacts with MHC-I groove residues while maintaining the anchor positions 2 and 8 (x[LM]xxxxx[ILV]) (Fig. 3a). Comparison of thermal stabilities ($T_m$) of HLA-A*02:01 refolded using N-terminal variants of TAX revealed a progressive reduction in $T_m$ as a result of destabilization of the peptide complex upon loss of N-terminal polar contacts (Fig. 3b). Both gTAX/HLA-A*02:01 and Ac-LLFGYPVYV/HLA-A*02:01, which gave the lowest $T_m$ values at 40 °C and 46 °C due to loss of polar contacts and steric clashes between the N-terminal acetyl and sidechains of groove residues (Fig. 3a), respectively, promoted the formation of peptide-deficient MHC-I/TAPBPR complexes in the presence of 10 mM GM dipeptide (Fig. 3c, f and Supplementary Fig. 2a) as shown by liquid chromatography-mass spectrometry (LC-MS) (Supplementary Fig. 2b). In contrast, the variant lLFGYPVYV, where the N-terminal L-Leu was substituted by the D-Leu enantiomer (l) leading to suboptimal packing interactions in the A-pocket (Fig. 3a) and consequently a reduced $T_m$ value of 54 °C (Fig. 3b), did not promote the formation of a stable complex with TAPBPR. We examined the stability of empty HLA-A*02:01/TAPBPR complexes by SDS-PAGE gel, to find that they remained intact for up to 6 months at −80 °C, without compromising their performance in peptide exchange reactions. Incubation with 10-fold molar excess of high-affinity TAX peptide induced dissociation of the complex, as

observed both by native gel and size-exclusion chromatography (SEC) assays (Fig. 3d and Supplementary Fig. 2c), with the loaded peptide detectable by LC-MS (Supplementary Fig. 2d). Accordingly, high-affinity peptides (TAX, CMVpp65 or MART-1) could be readily loaded into the empty complex, out competing fluorescently labelled TAMRA-TAX for HLA-A*02:01 binding (Fig. 3e). Using Bio-layer interferometry we measured the TAPBPR dissociation rate from HLA-A*02:01, which showed a significant increase in the presence of high-affinity peptides (TAX, CMVpp65 or MART-1), compared to an irrelevant peptide, used as a negative control (p29) (Fig. 3g).

Using the TAX/HLA-A*02:01 system as a benchmark we find that the overall yield of pMHC molecules prepared by TAPBPR exchange is approximately 2.5 times higher relative to the use of a photo-cleavable ligand[8], and approximately 10 times higher relative to the use of an empty HLA-A*02:01 molecule with an engineered disulfide bond[11] (Table 1). An additional benefit of our method is that the MHC-I/TAPBPR complexes are stable for several weeks at 4 °C, while their preparation and handling does not require dark conditions, as is the case for MHC-I molecules refolded with photo-cleavable ligands[8]. Using a similar approach, we designed suitable goldilocks peptides for the disease-relevant HLA-A*24:02 and HLA-A*68:02 alleles (Fig. 4a), and demonstrated exchange of the goldilocks for high-affinity peptides in the presence of TAPBPR by native gel (Fig. 4b) and by differential

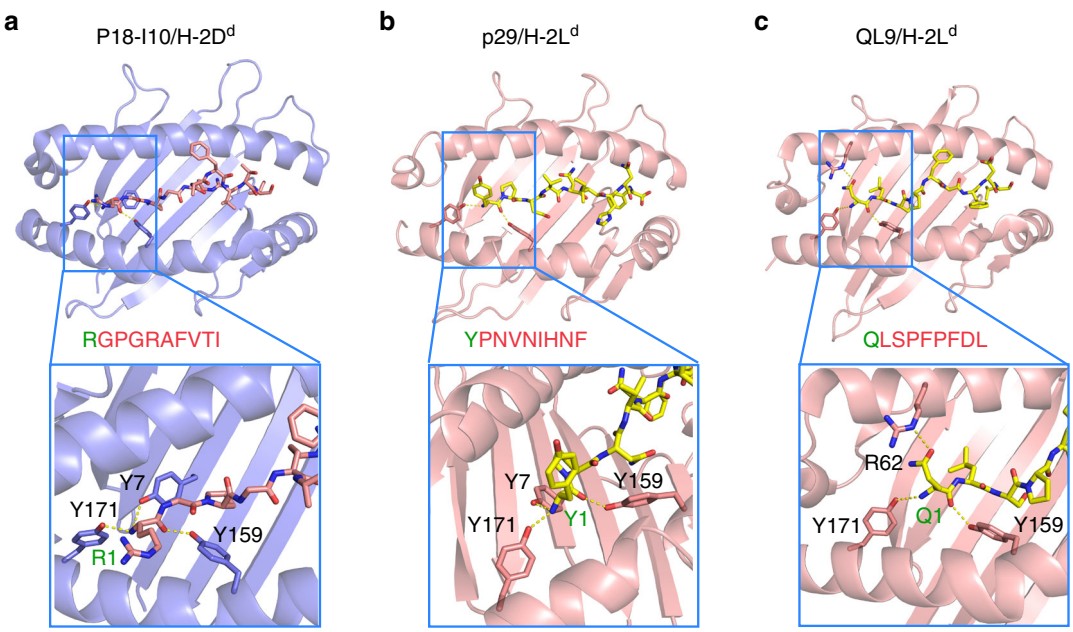

**Fig. 2 Structure-guided design of placeholder peptides for murine MHC-I molecules.** Crystal structures showing stabilizing contacts in the A-pocket of the MHC-I peptide-binding groove for: **a** P18-I10/H-2D$^d$ (PDB 3ECB[1]) with the bound peptide shown in light red. **b** Structure of p29/H-2L$^d$ (PDB 1LD9[2]), and QL9/H-2L$^d$ (PDB 3TF7[3]), with bound peptides shown in yellow. Insets focus on the A-pocket of the MHC-I peptide binding groove, with polar contacts between the N-terminal residue of the peptide and the indicated MHC-I residues shown as dotted yellow lines. Full peptide sequences are provided, with the first residue indicated in green.

scanning fluorimetry experiments (Fig. 4c). Several additional classical HLA alleles[21] are likely amenable to TAPBPR-mediated peptide exchange using our method.

**Loading high-affinity peptides using catalytic TAPBPR.** In some applications, purification of empty, stoichiometric MHC-I/TAPBPR complexes can be a limiting step. On the other hand, refolded goldilocks/MHC complexes can be aliquoted and stored at −80 °C for several months. To evaluate whether exchange of goldilocks for high-affinity peptides can occur when TAPBPR is offered in sub-stoichiometric concentrations, we used a native gel electrophoresis assay. This allowed us to monitor the formation of different pMHC species upon overnight incubation in the presence of 10-fold molar excess of different peptides and varying molar ratios of TAPBPR (Supplementary Fig. 3a). Under these conditions, complete peptide exchange on gTAX/HLA-A*02:01 was obtained using down to 1:1000 TAPBPR:MHC molar ratio, while no exchange was observed for a non-specific peptide, or in the absence of TAPBPR. In this assay the electrophoretic mobility of different pMHC molecules is dependent on the net charge of the bound peptide, which allowed us to resolve distinct protein bands of HLA-A*02:01 loaded with peptides of disparate charges from −2 to +1 in overnight catalytic peptide exchange reactions (Supplementary Fig. 3b). Taken together, these results highlight the use of TAPBPR as a peptide exchange catalyst, which can be advantageous for high-throughput applications.

**T cell staining using TAPBPR-exchanged pMHC molecules.** We then compared the performance of TAPBPR-exchanged phycoerythrin (PE)-tetramers, relative to PE-tetramers refolded in vitro using standard protocols[27]. Staining of a B4.2.3 TCR transgenic T cell line[28] with 1 μg/ml of TAPBPR-exchanged P18-I10/H-2D$^d$ tetramers prepared using either a stoichiometric (1:1) or catalytic (1:100) TAPBPR:MHC ratio, showed an identical population of tetramer positive cells relative to refolded pMHC-I tetramers. No staining was observed using an irrelevant motif

peptide (sequence AGPARAAAL), which binds with high affinity to H-2D$^d$ but is not recognized by the B4.2.3 TCR (Fig. 5a, b). The staining efficiency was further quantified using tetramer titrations, which showed a staining saturation curve with an EC$_{50}$ of 0.16 μg/mL compared to 0.44 μg/mL for refolded and TAPBPR-exchanged tetramers, respectively (Fig. 5c, with gating strategy shown in Supplementary Fig. 4). The 2.5-fold reduced staining efficiency can be attributed to sample loss due to the formation of aggregation prone, peptide-deficient H-2D$^d$ molecules during the overnight incubation with peptide. However, given that the observed EC$_{50}$ value remains well below the standard tetramer staining concentration of 1 μg/ml, TAPBPR exchange can still be used to obtain reliable staining results with the added benefit that, in contrast to refolding, TAPBPR exchange can be performed at high-throughput for a library of peptides (see below). A similar trend was observed in MART-1/HLA-A*02:01 tetramer staining of Jurkat/MA T cells transduced with a MART-1 specific TCR, DMF5 (Fig. 5a, b -middle), and in NY-ESO-1/HLA-A*02:01 tetramer staining of T cells transduced with a NY-ESO-1 specific TCR[29] (Fig. 5a, b -bottom). Here, TAPBPR-exchanged tetramers showed identical EC$_{50}$ values relative to refolded tetramers, highlighting the robustness of TAPBPR exchange for the preparation of human pMHC-I molecules in a high-throughput setting (Fig. 5c middle and bottom). No staining was observed using tetramers prepared by exchanging the mismatched peptides, NY-ESO-1 and MART-1 on T cells expressing the DMF5 and NY-ESO-1 TCRs, respectively (Fig. 5a, b).

**Monitoring peptide loading and TAPBPR release.** Since TAPBPR has been shown to promote the release of high-affinity peptides from the MHC groove in vitro[16], the persistence of TAPBPR following exchange may partially regenerate empty MHC-I molecules, thereby reducing the staining efficiency of the resulting tetramers. In a library format this may also lead to scrambling of excess peptides between MHC tetramers upon

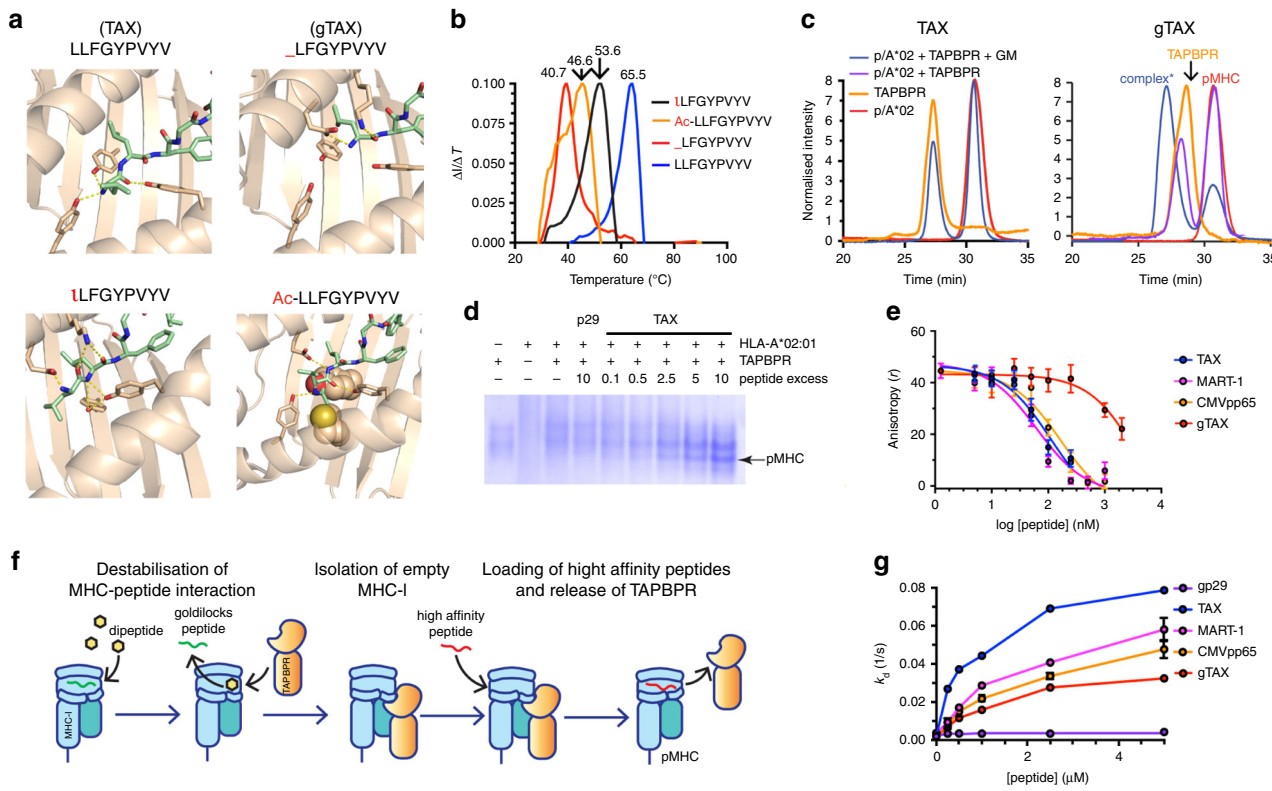

**Fig. 3 Capturing empty HLA-A*02:01/TAPBPR complexes for peptide exchange. a** Structure-based design of goldilocks peptides: comparison of polar contacts between HLA-A*02:01 and the N-terminal region of LLFGYPVYV (TAX) peptide (upper left). _LFGYPVPYV (gTAX) (upper right), Ac-LLFGYPVYV (bottom right) and lLFGYPVYV where l = D-Leucine (bottom left). Structures were modeled using PDB ID 1DUZ[26]. **b** Peptide complex thermal stabilities of HLA-A*02:01 bound to TAX, lLFGYPVYV, Ac-LLFGYPVYV and gTAX. **c** SEC TAPBPR binding assays of TAX/HLA-A02:01 (left), gTAX/HLA-A02:01 (right). **d** Native gel electrophoresis of HLA-A*02:01/TAPBPR complex incubated with a 10-fold molar excess of a non-specific peptide (p29, YPNVNIHNF) or varying molar excess of a specific, high-affinity peptide (TAX). 12% polyacrylamide native gels were run at 90 V for 5 h at 4 °C before visualization with InstantBlue (Expedeon). Data shown are representative of triplicate gel assays. **e** Competitive binding of TAMRA-TAX to purified HLA-A*02:01/TAPBPR complexes from (**c**) as a function of increasing peptide concentration, measured by fluorescence polarization. **f** Conceptual diagram of TAPBPR-mediated capture and peptide loading on empty MHC-I molecules. **g** Bio-Layer Interferometry analysis of TAPBPR dissociation from HLA-A*02:01 in the presence of peptides of different affinities. Data shown in **e** and **g** are representative of triplicate assays ($n = 3$) and error-bars are standard deviations from the mean. Source data are provided as a Source Data file.

**Table 1 Comparison of efficiencies between different peptide exchange methods.**

|  | Photosensitive peptide[7] | Disulfide mutant[11] | Goldilocks peptide |
|---|---|---|---|
| Yield per 1 L of refolding solution[a,b] | 4.0 ± 0.5 mg | 0.5 ± 0.2 mg | 5.0 ± 1.5 mg |
| Peptide exchange efficiency | 30–40% | 80–90% | 80–90% |
| Total pMHC yield | 1.6 mg | 0.4 mg | 4 mg |

[a]The proteins were refolded, concentrated and purified with size-exclusion and ion exchange chromatography.
[b]Data shown are representative of triplicate assays and error bars are standard deviation from the mean.

mixing, which would limit the use of molecular indices or barcodes to label tetramers according to their displayed peptides (see below). Specifically, the presence of TAPBPR induced the exchange of CMVpp65 loaded on HLA-A*02:01 tetramers for free MART-1 peptide, which can be detected by staining of DMF5+ T cells used here as a reporter (Supplementary Fig. 5b). To prevent this, full removal of TAPBPR molecules was readily achieved by spin column dialysis, immediately following the tetramerization and peptide loading steps as confirmed by SDS-PAGE (Fig. 1 and Supplementary Fig. 5a). The resulting pMHC-I tetramers did not capture free high-affinity peptides, even when offered at high (20×) molar excess, and peptide exchange between tetramers was undetectable by flow cytometry (Supplementary Fig. 5b, c).

Complete binding of high-affinity peptides on empty, tetramerized MHC-I/TAPBPR complexes can be monitored using a simple SDS-PAGE assay, through the disappearance of a distinct TAPBPR protein band following the spin column dialysis step. Here, TAPBPR is used in a similar manner to a conformation-specific antibody to label empty MHC-I molecules present after the peptide exchange reaction. Using this assay, we found that TAPBPR remained present on H-2D[d] tetramers following 1 h incubation with P18-I10 peptide, despite exhaustive spin column dialysis, due to the slow binding of peptide on empty H-2D[d]/TAPBPR molecules[16] (Supplementary Fig. 6a). As a result, H-2D[d] tetramers prepared upon incomplete loading with peptide show a 2.5 times higher $EC_{50}$ value relative to those obtained after complete peptide exchange, which can be readily achieved in an

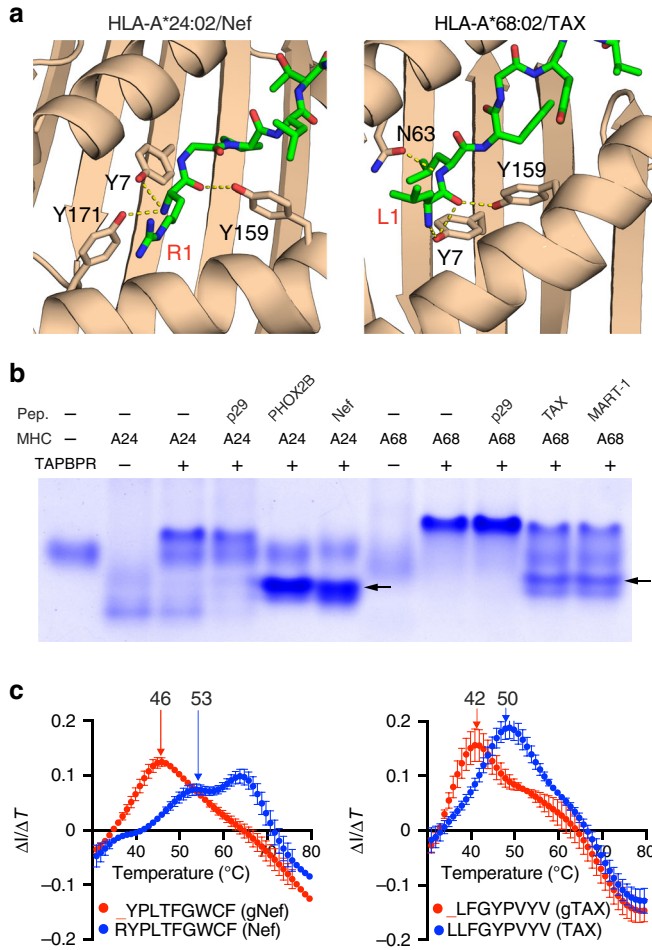

**Fig. 4 TAPBPR-mediated peptide exchange on HLA-A\*24:02 and HLA-A\*68:02. a** Structure-based design of goldilocks peptides: analysis of A-pocket hydrogen bonds observed in the X-ray structure of HLA-A\*24:02/Nef$_{134-10}$ (RYPLTFGWCF) and homology-based model of HLA-A\*28:02/TAX (LLFGYPVYV). Peptide residues are labelled in red, A-pocket MHC residues in black. The structure of HLA-A\*24:02/Nef$_{134-10}$ was obtained from PDB ID 3QZW[41]. The structure of HLA-A\*68:02/TAX was modeled using a related structure (PDB ID 4HX1[42]) as input. **b** Native gel electrophoresis analysis showing MHC-I/TAPBPR complex dissociation in the presence of 10-fold molar excess of relevant, high-affinity peptides PHOX2B (QYNPIRTTF), Nef$_{134-10}$ (RYPLTFGWCF) (lanes 5 and 6) and TAX, MART-1 (lanes 10 and 11) for complexes prepared using refolded HLA-A\*24:02 and HLA-A\*68:02 with gNEF (lane 2) and gTAX (lane 7) goldilocks peptides, respectively (protein yields of approximately 6 and 8 mg from a 1 L refolding reaction). The exchanged pMHC molecular species on lanes 5,6,10 and 11 are indicated with arrows, showing electrophoretic mobilities that depend on the charge of the bound peptide. Both complexes remain bound in the presence of 10-fold molar excess of the irrelevant peptide p29 (YPNVNIHNF) (lanes 4 and 9). 12% polyacrylamide native gels were run at 90 V for 5 h at 4 °C before visualization with InstantBlue (Expedeon). Data shown are representative of triplicate gel assays. All protein samples used in **a** and **b** were derived from the same peptide exchange experiment, and the gels were processed in parallel. **c** Overlaid Differential Scanning Fluorimetry temperature profiles of: goldilocks/MHC-I (red) and high-affinity peptide/MHC-I (blue) prepared using chaperone-mediated exchange of the goldilocks for high-affinity peptides, followed by purification of the pMHC peak by SEC. Thermal stabilities are shown: HLA-A\*24:02 refolded with gNEF (YPLTFGWCF) or exchanged with NEF (RYPLTFGWCF) (left), and HLA-A\*68:02 refolded with gTAX (_LFGYPVYV) or exchanged with TAX (LLFGYPVYV) (right). The peaks at approximately 63 °C correspond to the thermal melt of the $\beta_2$m light chain[37]. Data shown are representative of triplicate assays and error-bars are standard deviation from the mean. Source data are provided as a Source Data file.

overnight peptide loading reaction, as confirmed by our gel assay (Supplementary Fig. 6b). Thus, an overnight incubation with peptide is a requirement for TAPBPR-mediated exchange on the murine MHC-I molecules tested here.

**Fine-tuning TAPBPR interactions through α₃ domain mutants.**
We next sought to fine-tune TAPBPR interactions with murine MHC-I molecules previously shown to form stable complexes with TAPBPR even when bound to high-affinity peptides[16], an effect which limits their use towards high-throughput tetramer library production due to persistent peptide exchange activity in the pooled library. Towards reducing the affinity of TAPBPR for murine molecules, we explored designed mutations at the α₃ domain of the heavy chain which participates in direct interactions with TAPBPR, and does not contribute to the formation of the peptide-binding groove. Specifically, Met 228 is located at an edge loop of the α₃ immunoglobulin fold and forms a hydrophobic contact with TAPBPR residues in the X-ray structure of the H-2D$^d$/TAPBPR complex[17] (Fig. 6a, b). We hypothesized that a mutation at this position from a Met, present in H-2D$^d$ and H-2L$^d$, to a polar Thr residue, present in HLA-A\*02:01, would lead to a reduced binding affinity of peptide-bound MHC-I molecules for TAPBPR. In contrast to the WT molecule, H-2D$^d$M228T did not bind TAPBPR when the high-affinity P18-I10 peptide was present in the MHC-I groove. Notably, TAPBPR maintained binding to empty H-2D$^d$M228T upon dissociation of the goldilocks gP18-I10 peptide, to generate a peptide-receptive complex (Fig. 6c, d). These results highlight the requirements of a system that is amenable to large-scale tetramer library production using our approach: (i) formation of a stable TAPBPR complex with an

empty molecule and (ii) TAPBPR dissociation upon binding of high-affinity peptides to the MHC-I groove.

**Labeling individual peptide specificities with DNA barcodes.**
To test our method on a high-throughput setting we generated two HLA-A\*02:01 tetramer libraries, one containing 29 tumor epitopes identified in neuroblastoma tissues and another encompassing a range of 31 viral, neoantigen and autoimmune epitopes (Supplementary Tables 1 and 2). To link pMHC specificities with TCR V(D)J sequences present in polyclonal samples, we barcoded fluorophore-labelled pMHC tetramers, prepared using TAPBPR exchange, with biotinylated DNA oligonucleotides (oligos)[30] at a 1:1 molar ratio (DNA oligos: pMHC). Using tetramer staining titrations on our DMF5 reporter cells, we found that the barcoded tetramers show a similar $EC_{50}$ value relative to the non-barcoded control (Supplementary Fig. 7). We used an in-house oligo design compatible with 10x Genomics gel bead oligos in the 5′ V(D)J product (Supplementary Fig. 8a) adding an additional modality of cellular information to our recently described ECCITE-seq method[22]. This method incorporates a cellular barcode into cDNA generated from both tetramer oligos and cellular mRNAs, including TCR transcripts, thus connecting TCR sequences and other mRNAs with tetramer specificities. Formation of stable pMHC species upon loading of each peptide can be detected at high throughput using two complementary differential scanning Fluorimetry (DSF) assays (Supplementary Fig. 9). Together with the SDS-PAGE (Supplementary Figs. 5a, 6a) and fluorescence/Bio-layer interferometry assays (Fig. 3e, g and Supplementary

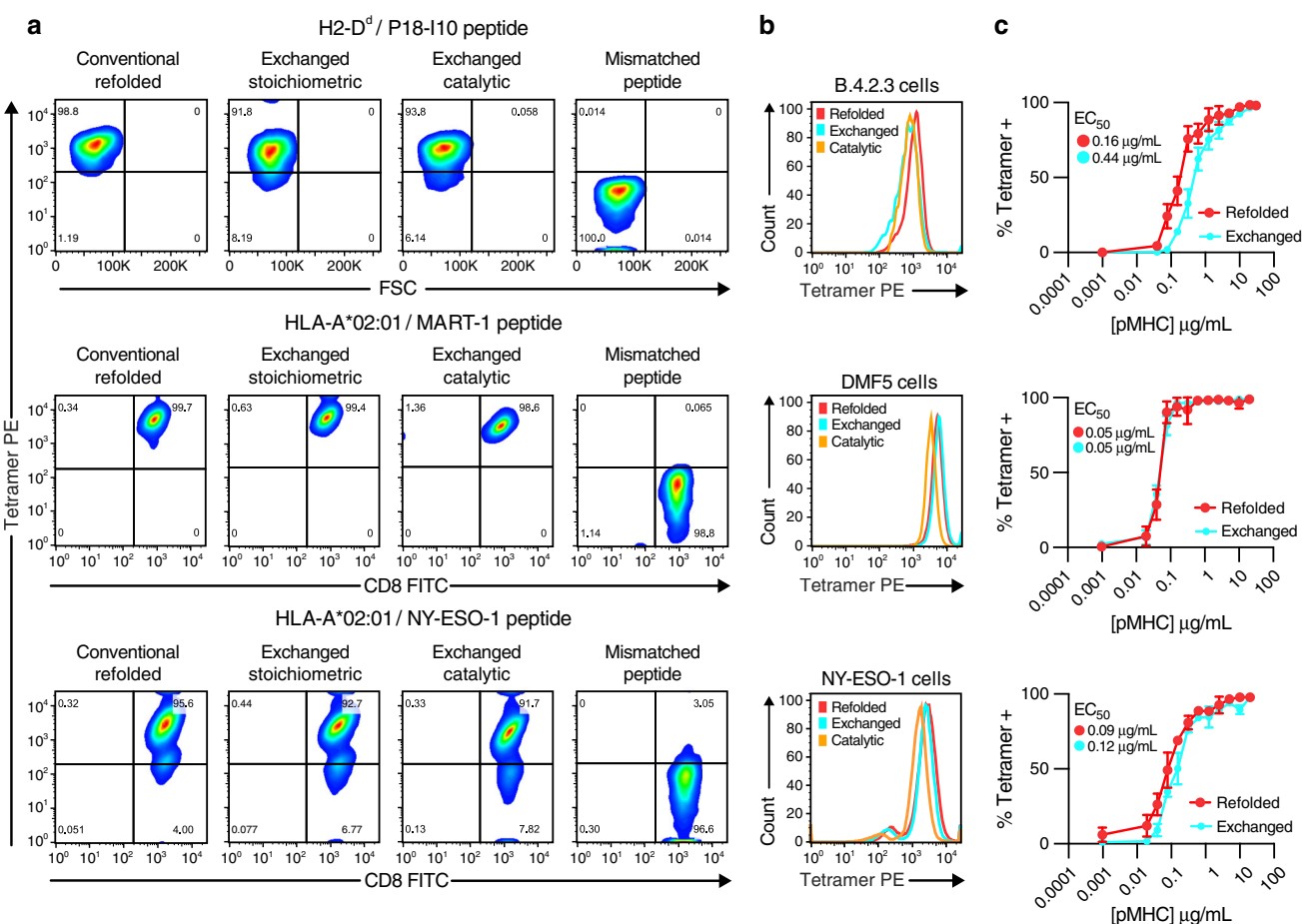

**Fig. 5 Flow cytometry using TAPBPR-exchanged pMHC-I tetramers. a** Representative flow cytometric analysis of top row; murine cells expressing the B4.3.2 TCR, middle row; DMF5 human T cells expressing the MART-1 TCR, bottom row; NY-ESO-1 human T cells expressing the NY-ESO-1 TCR. Columns depict staining with a fixed excess (1 μg/mL) of P18-I10/H2-D$^d$, MART-1/HLA-A*02:01 and NY-ESO-1/HLA-A*02:01 tetramers prepared either by conventional refolding, exchange of peptide on stoichiometric MHC-I/TAPBPR complexes, TAPBPR-mediated peptide exchange using a catalytic (1:100) TAPBPR to MHC-I molar ratio or by stoichiometric exchange of a mismatched peptide, not recognized by the respective TCRs. **b** Histogram plots of tetramer staining. **c** Tetramer titration of P18-I10/H-2D$^d$ (top), MART-1/HLA-A*02:01(middle) and NY-ESO-1/HLA-A*02:01 (bottom), prepared by exchange of each peptide on the corresponding stoichiometric MHC-I/TAPBPR complexes. Percentage of cells staining positive with tetramer over a serial two-fold dilution series were plotted and EC$_{50}$ values calculated by curve fitting to a sigmoidal line (with $R^2$ values in the 0.97–0.99 range), using Graph Pad Prism version 8 (GraphPad Software, La Jolla California USA). Data shown are representative of triplicate assays ($n = 3$) and error-bars are standard deviations from the mean. Gating strategies used for sorting tetramer-positive cells are outlined in Supplementary Fig. 4. Source data are provided as a Source Data file.

Fig. 10) the DSF data provide an additional means to confirm binding of a candidate peptide to the empty MHC-I groove, and can be performed on a 96-well format. After exhaustive dialysis of TAPBPR, excess barcodes and peptides, all individual peptide loading reactions were pooled into a single tetramer library sample for staining. Both final libraries prepared in this manner were further validated for: (i) the presence of all barcodes, using bulk sequencing reactions (Supplementary Fig. 8b) and (ii) staining of Jurkat/MA cells transduced with the DMF5 receptor specific to the MART-1 reference peptide included in each library. Here, the observed signal intensities ($10^3$–$10^4$) were sufficient to distinguish the approximately 33% population of DMF5 positive cells from the negative fraction (Supplementary Fig. 8c).

**Probing polyclonal TCR repertoires using pMHC libraries**. To confirm that our approach has sufficient sensitivity to detect antigen-specific T cells within a heterogeneous sample, 1% DMF5 Jurkat/MA T cells were spiked into a sample of CD8-enriched PBMCs co-cultured with dendritic cells and stained with the

library of 31 neoepitopes, including the MART-1 peptide. 100,000 tetramer positive cells were collected and 3,000 were sequenced. From this sparse sample, 256 cells with the highest number of DMF5 TCR reads were extracted and analyzed together with their corresponding MART-1 tetramer reads. In total, we recovered 85 cells with ≥10 MART-1 tetramer counts, 76 of which showed ≥10 DMF5 TCR reads (considered DMF5 positive) giving an approximate false positive rate of 10.6% (Fig. 7a). Conversely, 6 DMF5 positive cells showed ≤10 MART-1 tetramer counts, resulting in a false negative rate of 7.9%. The low number of cells with significant tetramer-barcode reads could be a function of tetramer avidity, exacerbated by high dilutions through the 10x Genomics system prior to cDNA preparation. This analysis shows that the discrimination between antigen-specific and non-specific T cells present in a mixed sample can be accomplished on the basis of tetramer-barcode reads with high confidence ($p$-value < 0.0001 using a two-tailed Mann–Whitney test), suggesting that tetramer libraries prepared by TAPBPR-mediated exchange can be used to identify sparse populations of antigen-specific T cells within a heterogeneous sample.

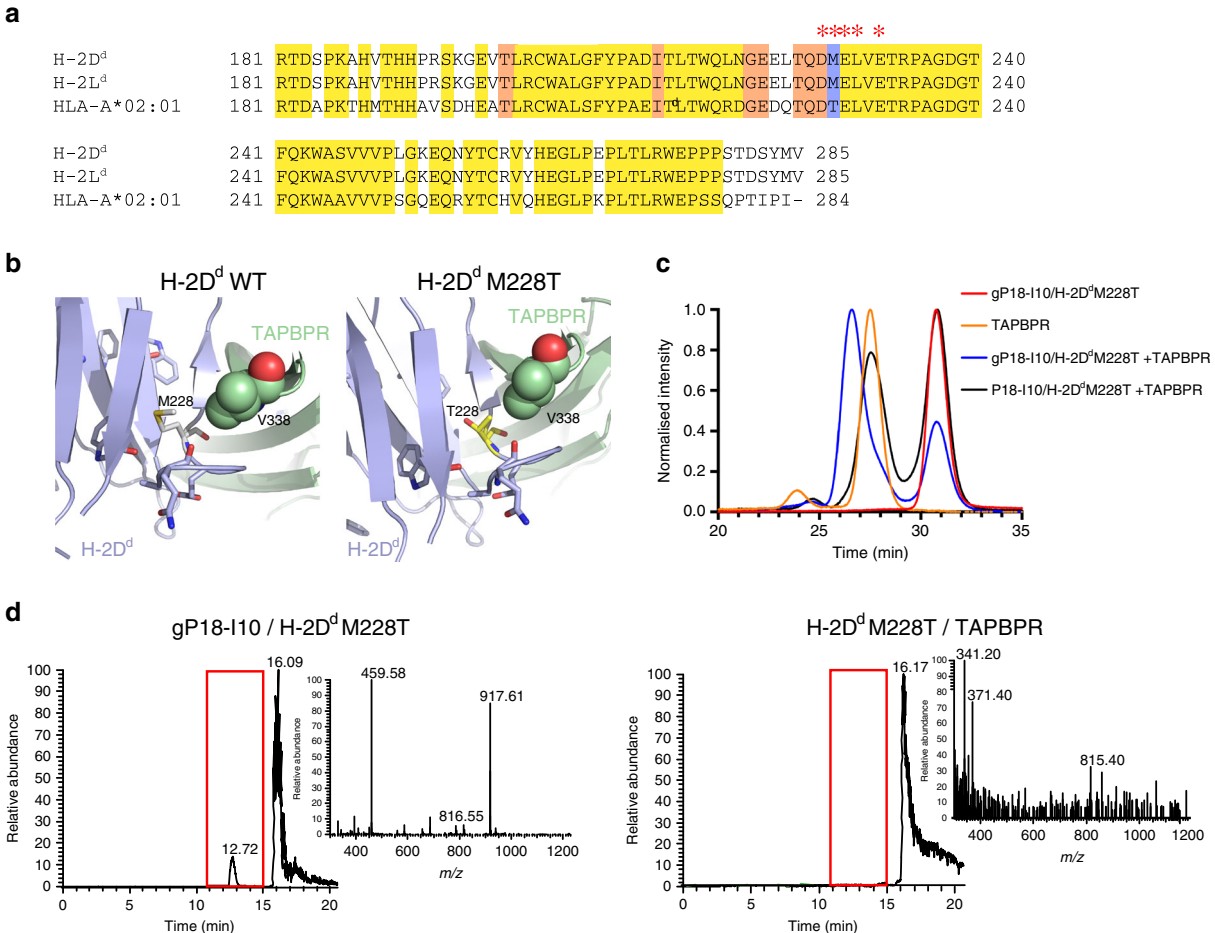

**Fig. 6 Fine-tuning MHC-I/TAPBPR interactions through α3 domain mutants. a** Alignment of the α$_3$ domain sequences from murine H-2D$^d$, H-2L$^d$ and human HLA-A*02:01. Conserved residues are highlighted in yellow. The M228T mutation site is highlighted in blue. * indicates residues directly participating in TAPBPR interactions, as shown in published mutagenesis studies and crystal structures. **b** TAPBPR/H-2D$^d$ α$_3$ domain interface from PDB ID 5WER[5] (left panel) and with the M228T mutation modeled (right panel). **c** Size exclusion chromatograms of H-2D$^d$ M228T refolded with either high-affinity P18-I10 or goldilocks gP18-I10 peptides, with and without TAPBPR. **d** LC/MS analysis of the peptide region from SEC-purified gP18/H-2D$^d$M228T and H-2D$^d$M228T/TAPBPR peaks from (**c**). Data shown are representative of triplicate SEC and LC/MS experiments.

Finally, we used our methodology to probe distinct TCR specificities present in a polyclonal repertoire of T cells. A total of 2 × 10$^6$ CD8+ T cells isolated from the spleen of an EBV-positive donor were stained using our barcoded tetramer library consisting of 34 viral, autoimmune and tumor epitopes (Supplementary Table 2). After sorting (gating strategy shown in Supplementary Fig. 11), 3,000 tetramer positive cells were loaded on the 10x platform for single-cell sequencing. ECCITE-seq analysis retrieved 1,722 cell barcodes, the majority of which were associated with <2 tetramer reads, giving a low apparent background. 110 cells were further identified as tetramer-enriched, defined as cells with >10 tetramer reads for at least one peptide specificity. 27 tetramer-enriched cells bound to >5 different peptide epitopes, 5 of which showed no bias towards a particular epitope and were excluded from subsequent analysis. Among the final set of 102 tetramer-enriched cells, we identified at total of 16 distinct epitope specificities, with an average of 200 reads per cell for each dominant tetramer (Fig. 7b). Specifically, a large fraction of tetramer-enriched cells had high reactivity for the NY-ESO-1 epitope (37 cells), followed by EBV-BRLF1 (16 cells), MAGE-A1 (10 cells) and IGRP$_{265-273}$ (7 cells). Towards validating the significance of our results, we focused on barcodes corresponding to the EBV-BRLF1 epitope (Fig. 7c) whose TCR repertoire has

been previously characterized using ad hoc tetramers prepared using conventional refolding protocols[31]. Inspection of V(D)J TCR sequence reads from tetramer-enriched cells identified a clear bias towards the usage of TRAV8-1 (50%) with TRAJ34 (50%), TRBV19 (21%), TRBJ2-1 (36%) and TRBJ2-7 (29%), a finding consistent with published reports on EBV-BRLF1 specific repertoires[32]. Whereas the β-chain CDR3 sequences were more variable, analysis of TRA CDR3 sequences further identified a known public α-chain CDR3 (CAVKDTDKLIF) which was previously linked to a functional TCR, specific for the EBV-BRLF1 epitope[31] (Fig. 7e). This sequence was observed in half of EBV-BRLF1 tetramer-enriched cells, and was not detected in cells enriched for any other tetramer. We also analyzed the TCR repertoire from 37 T cells with high barcode reads corresponding to the heteroclitic NY-ESO-1 epitope (SLLMWITQA). Here, the TCR α- and β-chain TCR sequences were more diverse, with some biases for TRBV28 (15%) and TRBJ2-7 (8%) usage (Fig. 7d). The observed diversity in TCR CDR3 regions that recognize NY-ESO-1 tetramers are consistent with the known high cross-reactivity of this epitope[33], and provide clues for engineering more exclusive TCRs for immunotherapy applications[34] (Fig. 7e). Taken together, these results corroborate our initial findings using the spiked DMF5/PBMC sample (Fig. 7a), and further suggest that our

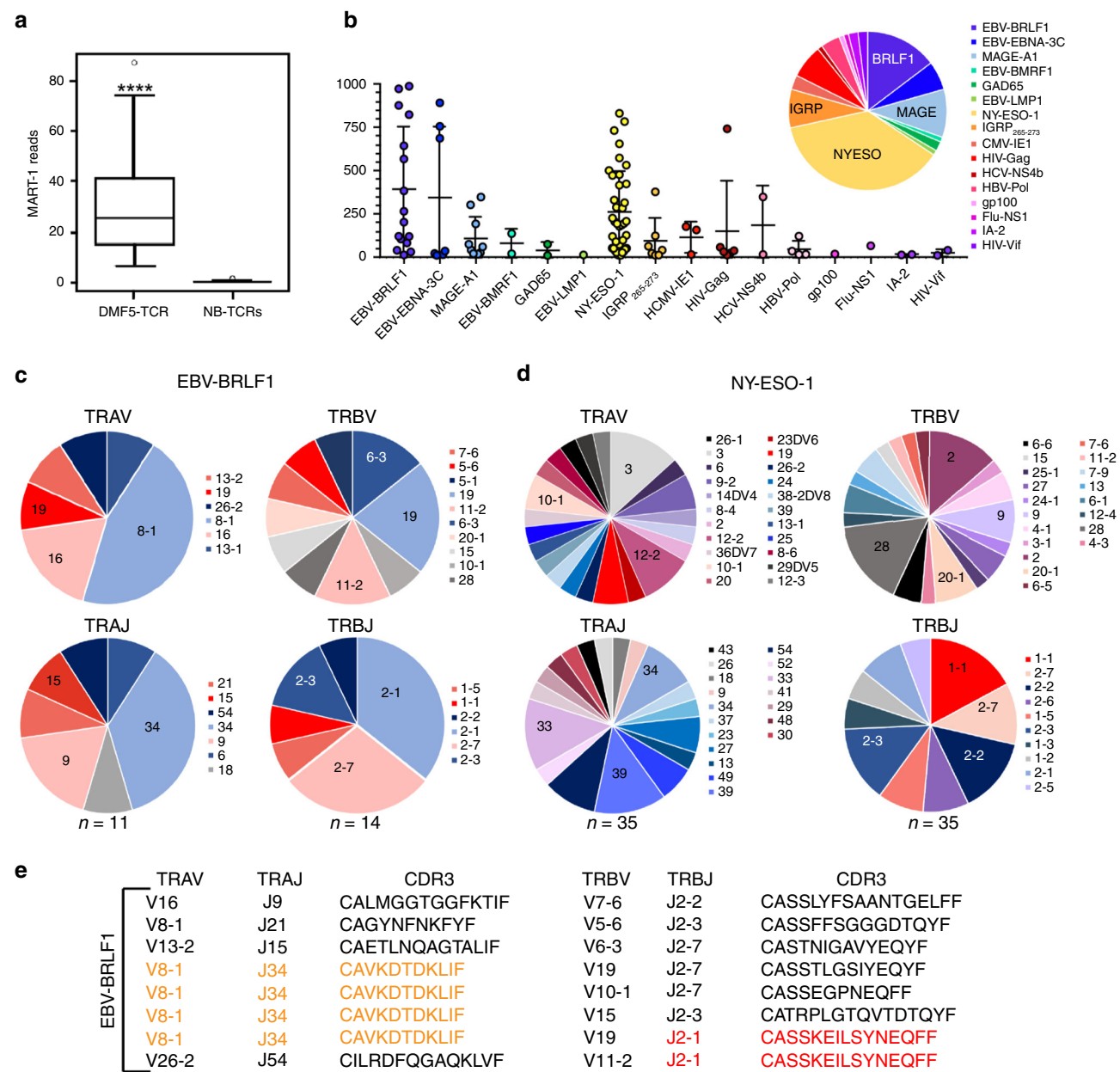

**Fig. 7 Identification of paired αβ TCR sequences with their antigen specificities. a** Recovery of MART-1 tetramer barcodes on DMF5 cells from PBMC-DC co-cultures spiked with 1% DMF5 cells. Number of MART-1 tetramer reads among DMF5 positive (DMF5-TCR) and negative (NB-TCR) cells. Cells are classified as positive/negative according to sequencing reads of the DMF5 TCR, where >10 DMF5 reads was used as a cutoff to classify DMF5+ cells. Box graphs display the distribution of MART-1 tetramer reads from DMF5+ (>10 DMF5 reads $n = 76$) and DMF5- ($\leq$10 DMF5 reads, $n = 927$) cells. Upper/ lower bars and box boundaries indicate the 95th/5th and 75th/25th percentiles, respectively, horizontal box lines indicate medians. Statistical significance was assessed using a two-tailed Mann-Whitney test ($p$-value < 0.0001). **b** Distribution of antigen specificities identified from tetramer + /CD8+ T cells from human splenocytes and the number of tetramer-barcode read per cell. Each dot represents a single cell, n = 102 in total, error bars correspond to one standard deviation from the mean. **c, d** V(D)J usage of cells identified as specific for the EBV-BRLF1 (YVLDHLIVV) and NY-ESO-1 (SLLMWITQA) antigens. All TRAVJ ($n = 11/35$) and TRBVJ ($n = 14/35$) chains identified are represented. **e** TCR CDR3 sequences identified for antigen-specific T cells. Consensus BRLF-1 TRA and NY-ESO-1 TRB chains are highlighted in orange and red, respectively. Gating strategies used for sorting tetramer+ cells are outlined in Supplementary Fig. 11. Source data are provided as a Source Data file.

barcoded pMHC libraries prepared using TAPBPR exchange are of sufficient quality and staining efficiency to identify antigen-specific TCRs present within polyclonal repertoires.

## Discussion

We have outlined a robust method to isolate stable, empty MHC-I molecules at milligram quantities, that can be readily loaded

with peptides of choice in a high-throughput manner. Rather than relying on chemically synthesized conditional ligands, our approach employs a molecular chaperone, TAPBPR, in an analogous manner to the antigen processing pathway used by cells to load MHC-I molecules with immunodominant peptides[1]. In combination with a simple indexing design that is compatible with the 10x Genomics system, our approach can efficiently link paired TCR V(D)J sequences and other transcription markers to

their pMHC specificities in a polyclonal sample setting. Our peptide exchange and barcode sequencing workflow has no theoretical upper limit with respect to library size, other than the cost of peptide and oligo synthesis, which renders it suitable for the simultaneous analysis of hundreds of epitope specificities in future experiments. In addition to expediting tetramer library preparation and the identification of novel TCR specificities, our method can be readily extended to include the analysis of complete transcriptomes and surface proteins[22,23], thereby providing a toehold for functional studies of TCR:pMHC antigen recognition.

## Methods

**Peptide sequences**. All peptide sequences are given as standard single letter code. Peptides used for MHC refoldings and production of the neoepitope library were purchased from Genscript at 98% purity or as pepsets from Mimotopes and dissolved in 8.25% Acetonitrile, 25% DMSO, and 66.75% $H_2O$. Peptides containing modifications: TAMRA-TAX and GILGFVFXL (where X = 3-amino-3-(2-nitrophenyl)-propionic acid) were purchased from Biopeptek at 98% purity. lLFGYP-VYV and Ac-LLFGYPVYV were synthesized in house using standard FMOC chemistry. Peptide binding affinities were predicted using netMHCpan 4.0[35].

**In vitro refolding of pMHC molecules**. Plasmid DNA encoding the luminal domain of class I MHC (MHC-I) heavy chains H-2D$^d$, HLA-A*02:01, H-2L$^d$ and human $\beta_2$-microglobulin (h$\beta_2$m,) were provided by the tetramer facility (Emory University), and transformed into *Escherichia coli* BL21(*DE3*) (Novagen). MHC-I proteins were expressed in Luria-Broth media, and inclusion bodies (IBs) were purified using standard protocols[27]. In vitro refolding of pMHC-I molecules was performed by slowly diluting a 200 mg mixture of MHC-I and h$\beta_2$m at a 1:3 molar ratio over 24 h in refolding buffer (0.4 M L-Arginine, 100 mM Tris pH 8, 2 mM EDTA, 4.9 mM reduced glutathione, 0.57 mM oxidized glutathione) containing 10 mg of synthetic peptide purchased from Genscript at 98% purity at 4 °C. H-2D$^d$ heavy chain was refolded with RGPGRAFVTI (P18-I10) derived from HIV gp120[36] or _GPGRAFVTI (gP18-I10). H-2L$^d$ was refolded with _PNVNIHNF (gp29) or QLSPFPFDL (QL9) derived from oxo-2-gluterate dehydrogenase. HLA-A*02:01 was refolded with variants of LLFGYPVYV (TAX) derived from HTLV-1 including _LFGYPVYV (gTAX), N-terminally acetylated TAX (Ac-LLFGYPVYV), lLFGYPVYV where the first residue is a D-leucine or with ELAGIGILTV (MART-1) derived from Melan-A. Refolds were allowed to proceed for 96 h followed by size-exclusion chromatography (SEC) using a HiLoad 16/600 Superdex 75 column (150 mM NaCl, 25 mM Tris pH 8) at a flow rate of 1 mL/min, followed by anion exchange chromatography on a mono Q 5/50 GL column at 1 mL/min a 40 minute 0-100% gradient of buffer A (50 mM NaCl, 25 mM Tris pH 8) and buffer B (1 M NaCl, 25 mM Tris pH 8). Typical protein yields from a 1 L refold were 5–10 mg of purified pMHC-I.

**Recombinant TAPBPR expression and purification**. The luminal domain of TAPBPR was expressed using a stable *Drosophila* S2 cell line (Dr Kannan Natarajan, National Institutes of Health) induced with 1 mM CuSO$_4$ for 4 days and purified using affinity-based and size-exclusion chromatography[17]. Briefly, His$_6$-tagged TAPBPR was captured from the supernatant by affinity chromatography using high-density metal affinity agarose resin (ABT, Madrid). Eluted TAPBPR was further purified by size exclusion using a Superdex 200 10/300 increase column at a flow rate of 0.5 mL/min in 100 mM NaCl and 20 mM sodium phosphate pH 7.2.

**Size exclusion chromatography**. SEC analysis of MHC-I/TAPBPR interaction was performed by incubating 40 µM purified pMHC-I molecules with purified TAPBPR at a 1:1 molar ratio in 100 mM NaCl, 20 mM sodium phosphate pH 7.2 for 1 h at room temperature. Complexes were resolved on an Superdex 200 10/300 increase column (GE healthcare) at a flow rate of 0.5 mL/min in 100 mM NaCl and 20 mM sodium phosphate pH 7.2 at room temperature. MHC-I/TAPBPR complexes eluted at 26.5 min. In the case of H-2L$^d$ and HLA-A*02:01, 10 mM GF and GM were added respectively both to the initial incubation and to the running buffer during chromatography.

**LC-MS analysis**. Peptide occupancy of SEC-purified MHC-I was determined by HPLC separation on a Higgins PROTO300 C4 column (5 µm, 100 mm × 21 mm) followed by electrospray ionization performed on a Thermo Finnigan LC/MS/MS (LQT) instrument. Peptides were identified by extracting expected *m/z* ions from the chromatogram and deconvoluting the resulting spectrum in MagTran.

**Preparation of photo-exchanged pMHC-I**. H-2D$^d$ refolded with RGPGRAFJ*TI (photo-P18-I10) and HLA-A*02:01 refolded with GILGFVFJ*L, where J* is the photo-cleavable residue 3-amino-3-(2-nitrophenyl)-propionic acid, were UV-irradiated at 365 nm for 1 h in the presence of 20-fold molar excess peptide at room temperature. Reactions were iced for 1 h then centrifuged at 14, 000 rpm for 10 min

to remove aggregates. Photo-exchanged pMHC-I was then used for DSF analysis or tetramer preparation.

**Differential scanning fluorimetry**. To measure thermal stability of pMHC-I molecules[37], 2.5 µM of protein was mixed with 10 x Sypro Orange dye in matched buffers (20 mM sodium phosphate pH 7.2, 100 mM NaCl) in MicroAmp Fast 96 well plates (Applied Biosystems) at a final volume of 50 µL. DSF was performed using an Applied Biosystems ViiA qPCR machine with excitation and emission wavelengths at 470 nm and 569 nm respectively. Thermal stability was measured by increasing the temperature from 25 °C to 95 °C at a scan rate of 1 °C/min. Melting temperatures ($T_m$) were calculated in GraphPad Prism 7 by plotting the first derivative of each melt curve and taking the peak as the $T_m$ (Supplementary Fig. 9a). Determation of $T_m$ values of TAPBPR-exchanged molecules additionally required subtraction of the TAPBPR melt curve from the curve obtained for the complex, then calculating the first derivative. This procedure, on average, enhanced the $T_m$ values calculated for TAPBPR-exchanged pMHC-I molecules by 1.5 °C, compared to refolded and photo-exchanged pMHC-I molecules. All samples were analyzed in duplicate and the error is represented as the standard deviation of the duplicates analyzed independently.

**Bio-Layer Interferometry**. In each experiment, HIS1K biosensor tips (ForteBio) were first baselined in a buffer of 20 mM sodium phosphate pH 7.2, 100 mM NaCl and then coated with 6 µg/mL of HIS-Tagged TAPBPR in a matched buffer until the response was between 0.3 nm and 0.4 nm for each tip. TAPBPR coated biosensor tips were then dipped into matched buffer supplemented with 0.02% TWEEN-20 and 0.5 mg/mL BSA for 6 min to block non-specific interaction and as secondary baseline step. Subsequent steps were performed in the secondary baseline buffer (20 mM sodium phosphate pH 7.2, 100 mM NaCl, 0.02% TWEEN-20, 0.5% BSA). Biosensor tips were then dipped into buffer containing 10 µM HLA-A*02:01/g-TAX and 10 mM GM dipeptide to facilitate peptide deficient MHC-I/TAPBPR formation for 10 min. After peptide deficient MHC-I/TAPBPR formation on the biosensor tips, they were dipped into buffer supplemented with 0–5 µM of the indicated peptides for 14 min to facilitate pMHC dissociation from TAPBPR. Data was processed after subtraction of the reference sensor tip data set which was coated with TAPBPR in buffer, the *Y*-axis alignment to the secondary baseline, and an interstep correction alignment to dissociation. All data was locally fit for association and dissociation with a 2:1 (heterogeneous) binding model. Goodness of fit was determined according to R$^2$ values of above 0.99. All experiments were performed using an Octet Red 96e system and analyzed with the Octet data analysis HT v.11.1 software.

**Nano differential scanning fluorimetry**. In a volume of 20 µl, 1 µM peptide deficient HLA-A*02:01/TAPBPR was incubated with 20 µM of various peptides (Supplementary Table 2) in a buffer of 20 mM sodium phosphate pH 7.2, 100 mM NaCl, 0.02% TWEEN-20 for at least 1 h. Ten microliter of each sample was loaded on the Prometheus NT.48 instrument (NanoTemper) using the high sensitivity capillaries. NanoDSF measurements are performed using a temperature ramp rate of 1°C/min from 25 °C to 95 °C and an LED intensity of 20%. Data were analyzed using the PR Control software (NanoTemper). Melting temperatures (T$_m$ values) correspond to the inflection points of the first derivative of the 330/350 nm fluorescence ratio. Experiments were performed in duplicates and the shaded areas (Supplementary Fig. 9b) are representative of the error.

**Native gel electrophoresis**. Peptide-deficient MHC-I/TAPBPR complexes were incubated with the indicated molar ratio of relevant (TAX) or irrelevant (P18-I10) peptide for 1 h at room temperature. Samples were run at 90 V on 12 % polyacrylamide gels in 25 mM TRIS pH 8.8, 192 mM glycine, at 4°C for 4.5 h and developed using InstantBlue (Expedeon).

**Fluorescence anisotropy**. Fluorescence anisotropy was performed using TAX peptide labeled with TAMRA dye (K$_{TAMRA}$LFGYPVYV) (herein called TAMRA-TAX)[16]. Briefly, 50 nM of peptide-deficient HLA-A*02:01/TAPBPR complex in 100 mM NaCl, 20 mM sodium phosphate and 0.05 % (v) tween-20, were incubated with 0.75 nM TAMRA-TAX and graded concentrations of MART-1, CMVpp65 or unlabeled TAX peptide in total volumes of 100 µL in black 96 well assay plate (Costar 3915) for 2 h at room temperature while Fluorescence anisotropy (*r*) was recorded. Fluorescence anisotropy (FA) was recorded on a Perkin Elmer Envision 2103 with an excitation filter of $\lambda_{ex}$ = 531 nm and an emission filter of $\lambda_{em}$ = 595 nm. FA was normalized against TAMRA-TAX alone. Measurements were recorded every 30 seconds and data points represented are an average of FA values acquired following 105 min of incubation. All experiments are representative of at least 3 individual experiments run in triplicates. Data points were plotted and fit using a sigmoidal response curve in GraphPad Prism 7.

**Preparation of barcoded tetramers**. Purified, *BirA*-tagged pMHC-I molecules were biotinylated using the *BirA* biotin-protein ligase bulk reaction kit (Avidity), according to the manufacturer's instructions. Biotinylated pMHC-I was buffer exchanged into PBS pH 7.4 using Amicon Ultra centrifugal filter units with the

membrane cut-off 10 kDa. The level of biotinylation was evaluated by SDS-PAGE gel-shift assay in the presence of excess streptavidin.

In the stoichiometric approach, equimolar quantities of MHC-I loaded with goldilocks peptides and TAPBPR were added to streptavidin-PE (Prozyme) (2:1 MHC-I:streptavidin molar ratio) in ten additions every 10 min at room temperature. For tetramer barcoding, custom biotinylated DNA oligos (IDT) were used. Each tetramer was barcoded by adding 2:1 molar equivalent of DNA barcodes relative to streptavidin and incubated for 1 h, at room temperature. Peptide-deficient barcoded-MHC-I/TAPBPR tetramers were then exchanged with peptides of interest by adding a 10-fold molar excess of peptide to each well and incubating overnight at 4 °C. Additionally, 10-fold molar excess biotin (to block any free streptavidin sites) was added and incubated for a further 1 h at room temperature. After exchange, tetramers were transferred to Amicon Ultra centrifugal filter units with the membrane cut-off 100 kDa and washed with 1000 volumes of PBS to remove TAPBPR and excess of peptide and barcodes. After washing, exchanged tetramers were pooled and stored at 4 °C for up to 3 weeks. In the catalytic approach, the MHC-I loaded with goldilocks peptides were mixed with a 1:100 molar ratio of TAPBPR and added to streptavidin-PE (Prozyme) in the same way as described for the stoichiometric approach. All the further procedures were exactly the same.

**Cell culture.** 58 $\alpha^-\beta^-$ T cells expressing the B4.2.3 TCR, which recognizes P18-I10 bound to H-2D$^d$, were obtained from Dr. Kannan Natarajan (NIH). TCR β-chain deficient Jurkat-MA T cells expressing the DMF5 TCR, which recognizes Melan-A epitope MART-1 bound to HLA-A*02:01 and Jurkat-MA T cells expressing the NY-ESO-1 TCR which recognizes the NY-ESO-1 epitope bound to HLA-A*02:01 were generated as described below. All three lines were grown in DMEM supplemented with 10 % FBS, 25 mM HEPES pH 7, 2 μM β-mercaptoethanol, 2 mM L-glutamine, 100 U/mL penicillin/streptomycin and 1 x non-essential amino acids. Cells were maintained in exponential phase in a humidified incubator at 37 °C with 5% CO$_2$. Splenocytes from HLA-A*02:01+ organ donors were obtained through the Human Pancreas Analysis Program (http://hpap.pmacs.upenn.edu/) (University of Pennsylvania) after informed consent by each donor's legal representative.

**Generation of DMF5 and of NY-ESO-1 T cell lines.** Retrovirus for transduction of Jurkat/MA[38] and primary CD8 T cells was produced using Platinum-A retroviral packaging cell line. DMF5/NY-ESO-1 cassettes were assembled using previously described CDR3 sequences and V(D)J family genes[39], codon optimized, synthesized, and cloned into pMP71 retroviral vector[40]. Jurkat/MA cells were plated in 6-well plates at $7 \times 10^5$ cells/well and transfected with 2.5 mg of retroviral vector pMP71 using Lipofectamine 3000 (Life Technologies, Invitrogen). After 24 h, medium was replaced with IMDM-10 % FBS or AIM-V-10 % FBS. Supernatants were harvested and filtered with 0.2 mM filters after 24 h incubation and transferred to Jurkat/MA cells in 6-well plates pre-treated with 1 mL well/Retronectin (20 mg/mL in PBS, Takara Bio. Inc.,) at $1 \times 10^6$ cells/well and spinoculated with 2 mL of retroviral supernatant at 800 g for 30 min at RT. After 24 h, cells were washed and PBS, and cultured in IMDM-10% FBS. Jurkat/MA cells were stained with MART-1/ NY-ESO-1 dextramers (Immudex), and sorted for dextramer positive cells.

**PBMC/DC co-culture.** Normal donor monocytes were plated on day 1 in 6-well plates at $5 \times 10^6$/ well in RPMI-10 FBS supplemented with 10 ng/ml IL-4 (Peprotech) and 800 IU/ml GM-CSF (Peprotech) and incubated at 37 °C overnight. On day 2, fresh media supplemented with 10 ng/ml IL-4 and 1600 IU/ml GM-CSF was added to the monocytes and incubated at 37 °C for another 48 h. On day 4, non-adherent cells were removed and immature dendritic cells washed and pulsed with 5 μM peptide in AIM-V-10 % FBS supplemented with 10 ng/ml IL-4, 800 IU/ml GM-CSF, 10 ng/ml LPS (Sigma-Aldrich), and 100 IU/ml IFN-γ (Peprotech) at 37°C overnight. Day 1 was repeated on days 4 and 8 to generate dendritic cells for the second and third stimulations on days 8 and 12, respectively. On day 5, normal donor-matched CD8+ T cells were co-cultured with the pulsed dendritic cells in AIM-V-10 % FBS. Day 5 protocol was repeated on day 8 and day 12 using dendritic cells generated on days 4 and 8 for the second and third stimulation, respectively.

**Flow cytometry.** Tetramer analysis was carried out by staining $2 \times 10^5$ cells with anti-CD8α mAb (BD Biosciences) and 1 μg/mL of HLA-A02:01/MART-1, HLA-A02:01/NY-ESO-1 or 1 μg/mL H-2D$^d$/P18-I10 tetramer for 30 min on ice, followed by two washes with 30 volumes of FACS buffer (PBS, 1% BSA, 2 mM EDTA). Live/dead gating determined by staining with propidium iodide. All flow cytometric analysis was performed using a BD LSR II instrument equipped with FACSDiva software (BD Biosciences). For cell sorting experiments, cryopreserved human splenocytes were thawed and rested in RPMI media (10% FBS, 1% L-glutamine, 1% Pencillin/Streptomycin). CD8+ T cells were enriched by negative selection using magnetic beads according to the manufacturer's protocol (STEMCELL Technologies). Cells were then treated with dasatinib (50 nM, Sigma-Aldrich) for 30 min prior to staining. Afterward, 50 μL each of PE and APC versions of the tetramer library were added (final amount was 0.5 μg pMHC per tetramer) were added for 15 min at room temperature. Cells were washed and resuspended in BD pre-sort buffer (BD Biosciences). Cell sorting was performed on a FACS Aria FUSION (BD Biosciences). Live cells were gated based on forward and side scatter profiles and data was analyzed using FlowJo software (FlowJo, LLC). For EC$_{50}$ determination, tetramer concentrations were calculated based on total amount of pMHC-I at the time of exchange. Titrations were performed on the appropriate cell line in triplicate in two independent experiments. The percentage of tetramer-positive T cells was measured relative to the staining achieved at the highest concentration tested within each experiment. EC$_{50}$ values were calculated by fitting a Boltzmann sigmoidal function to the data GraphPad Prism 7.

**ECCITE-seq.** Post sorting, samples were prepped for the 10X Genomics 5 P V(D)J kit workflow, and processed according to the ECCITE-seq protocol[22], with the modifications below (Supplementary Fig. 8a). All primer oligonucleotide sequences referenced below are outlined in Supplementary Table 3. For cDNA amplification, 1 μL of 0.2 μM tetramer additive was spiked into the reaction. Post cDNA PCR, a 0.6x SPRI cleanup was performed, resulting in the larger cDNA fragments being retained on the beads, and the tetramer tags in the supernatant. After separation of the two fractions and elution from the beads, a portion of the cDNA was used to perform TCR α/β amplification and library prep, as described in the 10X genomics protocol. A separate portion of the cDNA elution was used to perform a DMF5 receptor specific enrichment, using a hemi-nested PCR strategy akin to that used for the TCRα/β enrichment. All PCRs were performed using 2X KAPA Hifi Master Mix (Kapa Biosystems), with the additional primers DMF5_PCR1,DMF5_PCR2, RPI-x primer (x nucleotides comprise a user-defined index) and P5_generic (Supplementary Table 3). The supernatant of the 0.6X SPRI purification in step 2 above was purified with 2 rounds of 2x SPRI. First, 1.4x SPRI was added to the supernatant to bring up the volume to 2×, followed by two rounds of 80% ethanol washes. After eluting in water, an additional 2x SPRI cleanup was performed. Post second cleanup, the tetramer tags were converted to a sequenceable library by PCR with the additional primer N7XX (x nucleotides comprise a user-defined index).

**Sequencing and analysis.** Individual tetramers were pooled in one library sample prior to sequencing. Samples were sequenced on a Miseq using a v2 300 cycle kit (151 cycles R1, 8 cycles I1, 151 cycles read 2). Post sequencing, TCR fastq files were pooled together for each sample, then analyzed using cellranger vdj 3.0.0 against the GRCh38 reference genome (v2.0.0, as provided by the 10X website). To identify the DMF5 receptor, we used CITE-seq-Count version 1.4.1 to search for the DMF5 specific tag, using default parameters (hamming distance set to 5). For tetramers, we used CITE-seq-Count version 1.4.1 using all default parameters, with the exception of hamming distance set to 1, and a whitelist to search for only cells with TCR found by 10×.

**TCR Repertoire Analysis.** All analysis was performed using the PE-tetramer barcodes alone. Cells with ≥10 tetramer reads were clustered into pMHC specificity groups based on the tetramer-barcode read. Cells with multiple tetramers with >10 reads were clustered based on the most frequent tetramer read (≥50% of total tetramer reads for that T cell). All TCR sequences identified (partial or complete) were used in global VJ usage analysis. In cases where multiple TCRs were read, only TCR sequences with the highest true reads were used (generally representative of ≥90% of TCR reads for that T cell). Known receptors and CDRs were queried and identified using VDJdb and literature searches.

**Reporting summary.** Further information on research design is available in the Nature Research Reporting Summary linked to this article.

## Data availability

Raw and processed Illumina sequencing data are deposited with National Center for Biotechnology Information's Gene Expression Omnibus under accession codes GSE146826, GSM4407564, GSM4407565, GSM4407566 and GSM4407567.

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

## Acknowledgements

The authors acknowledge Andrew McShan (UCSC), Marlon Stoeckius (NYGC) and Erik Procko (UIUC) for helpful discussions throughout the course of this study. We are grateful to Kannan Natarajan and David Margulies (NIH) for providing the TAPBPR S2 and B4.2.3 TCR mammalian cell lines, Alejandro Rodriguez and Jevgenij Raskatov for synthesis of Ac-LLFGYPVYV and lLFGYPVYV peptides, Brett Thurlow and Charles Heffern (nanoTemper), for recording and analyzing nanoDSF data. Work at NYGC was supported by grants to P.S. from the NIH (R21HG009748) and the Chan Zuckerberg Initiative (HCA-A-1704-01895). This work was also supported by grants from the National Institutes of Health to N.G.S. (R35GM125034, R01AI143997), to M.R.B. (UC4 DK112217), and to J.M.M. (R35CA220500, U54CA232568). We acknowledge the CIRM Shared Stem Cell Facility grant to UCSC (CL1-00506). This manuscript used data acquired from the Human Pancreas Analysis Program (HPAP-RRID:SCR_016202) Database (https://hpap.pmacs.upenn.edu), a Human Islet Research Network (RRID: SCR_014393) consortium (UC4-DK-112217 and UC4-DK-112232).

## Author contributions

N.G.S. conceived and designed the project. S.A.O., S.M.O., and G.I.M. produced and characterized murine MHC-I/TAPBPR complexes. S.A.O. and S.M.O. performed flow cytometry experiments. S.A.O., D.M., and G.I.M. produced recombinant TAPBPR protein. J.S.T., G.I.M., D.M., and N.G produced and characterized HLA/TAPBPR complexes, J.S.T., G.I.M., and N.G. performed Fluorescence anisotropy and bio-layer interferometry assays and made tetramer libraries. N.G.S., M.Y., and P.S. conceived and designed the binding specificity to TCR sequences with oligo barcoded tetramers, S.H. and P.S. performed sequencing experiments and curated sequence data. S.A.O., M.Y., and S.H. analyzed V(D)J sequencing data. M.Y., S.N., A.J., M.B., and J.M.M. designed experiments using polyclonal cell samples, provided all samples and performed flow cytometry and sorting experiments. S.A.O., S.M.O., and N.G.S. wrote the manuscript, with feedback from all authors.

## Competing interests

N.G.S. and P.S. are listed as inventor and co-inventor on patent applications related to the preparation of peptide-deficient MHC-I/Chaperone complexes and ECCITE-seq, respectively (US patent number WO 2020/010261 and provisional application 62/694-824). The other authors declare no competing interests.
