## [Peer Review File · Nature Communications]

Reviewers' comments:

Reviewer #1 (Remarks to the Author):

Sarah A. Overall et. al., "High Throughput 1 pMHC-I Tetramer Library Production Using Chaperone Mediated Peptide Exchange."

The goal of the manuscript is to use recent understandings of MHC-I peptide editing by TAPBPR to develop a method to exchange "high affinity" peptides into soluble MHC-I molecules, thereby allowing multiple pMHC tetramers to be created without the need to do parallel in vitro refolding reaction. Atop this method, current DNA barcoding methods were included, enabling the 10x genomics single cell RNAseq platform to pair T cells with their antigen specificity. Overall, the methodology itself appears to be fairly robust for tetramers generated from a collection of previously identified foreign (CMV) and auto (MART-1) antigens. My concerns predominately focus on journal appropriateness. I am a bit confused as to why this manuscript is at Nature Communications and not Nature Methods, or some other methods journal. There are obvious reasons (and needs) to develop new methodologies in all areas of science, however, in this manuscript there appears to be very limited (if any) new biological or chemical discovery science.

A few questions about the methodologies:

In Figure 2, it is unclear how well the peptide exchange process works. In particular, Figure. 1h and I suggest that refolded MHC-I molecules are ~3-10-fold more potent at staining, relative to TAPBPR exchanged peptides. Some understanding of this is important, either the original peptide is still present within some of the MHC-I molecules, or the heavy chain has disassociated from B2M. Both problems make generating competent tetramers problematic as they limit the number of possible functional MHC-I ligands were "tetramer." This problem becomes increasingly challenging when bio-DNA barcodes are also included in the "tetramer." E.g., a "tetramer" can quickly become a dimer (2 "good" MHC-I ligands + an unfolded HC + Bar code). The major source of this problem (empty MHC-I/unfolded HC, and MHC-I carrying the wrong peptide) largely do not occur in refolded preparations.

Also in figure 3, no statistics are shown. Just comments in the main text arguing for a positive correlation between DMF5 reads and MART-1 reads. By eye, I don't see this in 2a, unless one simply bins the data as 0 = 0, and >0 = >0.

Reviewer #2 (Remarks to the Author):

The manuscript by Overall and colleagues addresses an important aspect of immunology research, optimizing high throughput pMHC tetramer library. The group is excellent, and is clearly quite productive, although sometimes that makes it difficult to determine what is actually new in this report. At very least, it appears that the including of a human HLA-A2 is new, although it is somewhat unclear to me what aspects of the barcoding method are new here vs. previously reported. The manuscript is very well written. There are a few places where statements describing attributes of the system are not truly proven. For instance, they describe stability at room temperature or -80C for months, but don't give evidence to support this. I also am interested as to whether the two-fold enhancement in mean fluorescent intensity as compared to tetramers generated by UV exchange is truly relevant and have similar concerns for the 2.6 fold improvement in EC50 without differences in staining intensity. In

addition, it is unclear whether the metrics they show include the spin column dialysis that they describe needing to fully remove TAPBPR.

Reviewer #1 (Remarks to the Author):

Sarah A. Overall et. al., "High Throughput 1 pMHC-I Tetramer Library Production Using Chaperone Mediated Peptide Exchange."

The goal of the manuscript is to use recent understandings of MHC-I peptide editing by TAPBPR to develop a method to exchange "high affinity" peptides into soluble MHC-I molecules, thereby allowing multiple pMHC tetramers to be created without the need to do parallel in vitro refolding reaction. Atop this method, current DNA barcoding methods were included, enabling the 10x genomics single cell RNAseq platform to pair T cells with their antigen specificity. Overall, the methodology itself appears to be fairly robust for tetramers generated from a collection of previously identified foreign (CMV) and auto (MART-1) antigens. My concerns predominately focus on journal appropriateness. I am a bit confused as to why this manuscript is at Nature Communications and not Nature Methods, or some other methods journal. There are obvious reasons (and needs) to develop new methodologies in all areas of science, however, in this manuscript there appears to be very limited (if any) new biological or chemical discovery science.

We sincerely thank the reviewer for their positive appraisal of our work and their constructive comments. Regarding the appropriateness of Nature Communications as a journal choice, in fact our manuscript was originally submitted to *Nature Methods* in May, 2019. Upon communication between the Editors of the two journals, they decided that *Nat. Communications* would be a better fit for our work and therefore transferred our manuscript. We sincerely believe that this is the right audience, given the multi-disciplinary nature of our work which bridges biochemistry, single-cell transcriptomics with experimental immunology and its applications toward immunotherapy. Regarding new biological discovery included in the paper, our work outlines a proof-of-concept application to characterize the polyclonal TCR repertoire of an EBV positive donor, which corroborates previously published results and extends them further by providing new TCR V(D)J sequences specific for a common EBV epitope. In the revised manuscript we have placed particular attention in highlighting these points to address a general audience.

A few questions about the methodologies:

In Figure 2, it is unclear how well the peptide exchange process works. In particular, Figure 1h and I suggest that refolded MHC-I molecules are ~3-10-fold more potent at staining, relative to TAPBPR exchanged peptides. Some understanding of this is important, either the original peptide is still present within some of the MHC-I molecules, or the heavy chain has disassociated from B2M. Both problems make generating competent tetramers problematic as they limit the number of possible functional MHC-I ligands were "tetramer." This problem becomes increasingly challenging when bio-DNA barcodes are also included in the "tetramer." E.g., a "tetramer" can quickly become a dimer (2 "good" MHC-I ligands + an unfolded HC + Bar code). The major source of this problem (empty MHC-I/unfolded HC, and MHC-I carrying the wrong peptide) largely do not occur in refolded preparations.

The reviewer raises a valid concern regarding the presence of empty MHC molecules in our final tetramer preparation, which may in turn compromise the staining efficiency. Empty molecules would lead to a reduced number of "effective" pMHC molecules, which could become limiting when the avidity of the tetramers is further reduced by the presence of the DNA barcode (see, for example *Woolridge et al., Immunology, 126:147:164*). In our TAPBPR system, this effect is significantly more pronounced for tetramers prepared using the murine MHC-I molecule tested here (H2-D^d), whereas for the human MHC-I HLA-A*02:01 the TAPBPR-exchanged tetramers stain at equivalent intensity levels, and show a very similar titration curve to those prepared from refolded molecules (see updated Fig. 3 below).

The reviewer proposes the loss of β_2m or persistence of the original goldilocks peptide in some of the molecules as potential causes for this effect. While this is certainly a possibility, our results strongly suggest that, under our current experimental conditions, this is not the case. Specifically, we observe no detectable free β_2m in the sample by SEC or by native gel electrophoresis for all exchanged molecules tested here, either during the purification of the exchanged pMHC or upon storage of the tetramer for up to 2 months at 4 °C. Therefore, TAPBPR dissociation from the complex does not lead to release of the β_2m subunit. Likewise, the original goldilocks peptide is completely washed out from the sample during the SEC purification of the peptide-deficient MHC-I/TAPBPR complex (shown schematically in Fig. 2f). This is a key element of our methodology, which we have verified extensively by mass-spectroscopy. Specifically, the mass-spectra of purified MHC-I/TAPBPR complexes are completely lacking a peak corresponding to the mass of the gP18-I10 peptide for H2-D^d/TAPBPR (Supplementary Fig. 1d), and similarly of the gTAX peptide for HLA-A*02:01/TAPBPR (Supplementary Fig. 3b - right), whereas the same peptide masses can be readily observed in the initial goldilocks/MHC preparations (Supplementary Fig. 1c and 3b - left for gP18-I10/H2-D^d and gTAX/HLA-A*02:01, respectively).

An alternative explanation could be that the reduced staining of TAPBPR-exchanged H2-D^d tetramers after 1hr incubation with peptide is due to incomplete peptide loading and the presence of residual empty TAPBPR:MHC. To monitor such complexes after different time intervals of incubation with high-affinity peptide, followed by dialysis to remove the unbound peptide and TAPBPR, we examined the resulting tetramers by SDS-PAGE. Since the high-MW tetramerized MHC fraction does not enter the gel, we aimed to detect the presence of an approx. 50 kDa band corresponding to residual TAPBPR which binds with high-affinity to empty MHC molecules and can be therefore used in a similar manner to a "conformation-specific" Ab. For tetramers prepared for the human allele, HLA-A*02, TAPBPR is completely removed during the dialysis step (Supplementary Fig. 6a), suggesting that all MHC molecules in the sample are bound to peptide, which is known to promote dissociation of TAPBPR from the peptide-HLA-A*02 complex (*Morozov et al., PNAS, 2015*). However, for the H-2D^d allele molecule, TAPBPR remained present in the tetramerized sample when peptide was not added in the TAPBPR:MHC complex, or even upon 1 hr incubation with the P18-I10 peptide.

Supplementary Figure 7: Efficiency of TAPBPR-mediated loading of a high affinity peptide on H2-D^d.

To remediate this issue, we examined longer incubation times to accomplish complete peptide loading and TAPBPR release. We find that this can be achieved in an overnight peptide loading reaction (Supplementary Fig. 7a), leading to a 3-fold improvement in the EC_{50} values of the resulting P18-I10/H-2D^d tetramers (Supplementary Fig. 7b). As a final validation of the efficiency of H2-D^d and HLA-A*02 tetramers prepared following the complete dissociation of TAPBPR, we performed a detailed comparison with refolded tetramers in new titrations on 3 cell lines expressing a unique T cell receptor (Fig. 3c):

Figure 3. Flow cytometry analysis of cells expressing unique TCRs using tetramers prepared by conventional refolding or TAPBPR-mediated exchange

As expected, both TAPBPR-exchanged tetramers prepared for HLA-A*02:01 in complex with the MART-1 and NY-ESO-1 peptides stain with equivalent EC_{50} values as tetramers prepared from molecules refolded together with the same peptides. The observed 2.5-fold reduced EC_{50} of exchanged P18-I10/H2-D^d tetramers relative to the refolded preparation, even when TAPBPR has been fully removed, can be attributed to sample loss due to the formation of aggregation prone, peptide-deficient H-2D^d molecules during the exchange reaction. However, given that the observed EC_{50} value remains well below the standard tetramer staining concentration of 1 μ g/ml, TAPBPR exchange can still be used to obtain reliable staining results. In contrast to refolding preparations, TAPBPR-mediated exchange can be performed at high-throughput for 100s of peptides with low cost. These important points and have been outlined in detail points have been included in the revised text (p 6-7). We sincerely thank the reviewer for raising this concern, as it led to a significant technical improvement of our method.

Also in figure 3, no statistics are shown. Just comments in the main text arguing for a positive correlation between DMF5 reads and MART-1 reads. By eye, I don't see this in 2a, unless one simply bins the data as $0 = 0$, and $>0 = >0$.

We thank the reviewer for their suggestion. In the revised manuscript, we have included a rigorous statistical analysis using a two-tailed Mann-Whitney test (Fig. 4a). This analysis shows that the discrimination between antigen-specific and non-specific T cells present in a mixed sample can be accomplished on the basis of tetramer barcode reads, with high confidence (p-value < 0.0001). This is further described in the main text (p 10, pp 2).

Reviewer #2 (Remarks to the Author):

The manuscript by Overall and colleagues addresses an important aspect of immunology research, optimizing high throughput pMHC tetramer library. The group is excellent, and is clearly quite productive, although sometimes that makes it difficult to determine what is actually new in this report. At very least, it appears that the including of a human HLA-A2 is new, although it is somewhat unclear to me what aspects of the barcoding method are new here vs. previously reported.

We kindly thank the reviewer for their remarks on the general productivity of our group, and the importance of the aspect addressed by our present work. We understand that there has been a misunderstanding about the broader implications of this study and overall novelty that has been lost in the highly technical aspects of the paper. In the revised manuscript we have placed particular emphasis in highlighting the novel aspects of our results, and placing them in the context of: 1) the currently used method for high-throughput peptide exchange using photo-cleavable conditional peptides (*Bakker et al., PNAS, 2008*) and 2) the most recent breakthrough in the literature which was published while our present paper was under review, and involves "empty" molecules with an engineered disulfide bond (*Saini et al., Science Immunol., 2019*). We present new results which are summarized in two main text and three supporting figures, along with extensive additions and revisions in the main text. We hope that these changes together will address the reviewers' concerns about novelty, improvement relative to existing methods and broader significance for immunology research.

The manuscript is very well written. There are a few places where statements describing attributes of the system are not truly proven. For instance, they describe stability at room temperature or -80C for months, but don't give evidence to support this.

We have performed a qualitative time course evaluation of the stability of purified TAPBPR:MHC complexes at -80 °C by SDS-PAGE gel, to find that they remained intact for up to 6 months without compromising their performance in peptide exchange reactions, or the staining efficiency of the resulting pMHC tetramers. Similarly, TAPBPR:MHC complexes remain stable at 4 °C for 4 weeks, after which there is appreciable sample loss due to degradation. These results have been included in the revised text (p 6, pp 1).

I also am interested as to whether the two-fold enhancement in mean fluorescent intensity as compared to tetramers generated by UV exchange is truly relevant and have similar concerns for the 2.6 fold improvement in EC50 without differences in staining intensity.

The reviewer raises a valid concern regarding the practical utility of our method relative to the current approach involving UV exchange. To address this, we have performed a detailed comparison of the two methods. The new results, shown in Supplementary Table 1 below, indicate that our method outperforms the UV exchange approach (Bakker *et al.*, *PNAS*, 2008) by a factor of 2.5 in terms of peptide exchange efficiency due to significant precipitation during the UV reaction, while also allowing for a significantly more convenient workflow. Specifically, the UV exchange approach involves the use of expensive, custom made photo-cleavable peptides (p*). Moreover, because these peptides are sensitive to UV light, the refolding, dialysis and purification of p*MHC complexes must be performed in the dark. Finally, in a library setting the peptide exchange reactions must be performed for each peptide individually prior to tetramerization, due to fact that the commercially available streptavidin fluorophore conjugates used for tetramerization are sensitive to UV light. Since our method does not rely on photochemistry, it also does not have these disadvantages and can therefore be readily used to generate pMHC libraries in a high-throughput manner.

Table 1. Comparison of refolding and peptide exchange efficiencies for three high-throughput methods.

	WT A2 WITH PHOTSENSITIVE PEPTIDE ¹	DISULFIDE A2 MUTANT ²	WT A2 WITH GOLDBLOCKS PEPTIDE ³
Yield per 1 L of refolding solution**	4.0 ± 0.5 mg	0.5 ± 0.2 mg	5.0 ± 1.5 mg
Peptide exchange efficiency	30-40%	80-90%	80-90%
Total pMHC yield	1.6 mg	0.4 mg	4 mg

* The proteins were refolded, concentrated and purified with size-exclusion and ion exchange chromatography

** Data shown is representative of triplicate assays and error-bars are standard deviation from the mean.

1 Toebes, M. *et al.* Design and use of conditional MHC class I ligands. *Nat Med* **12**, 246-251, doi:10.1038/nm1360 (2006).

2 Saini, S. K. *et al.* Empty peptide-receptive MHC class I molecules for efficient detection of antigen-specific T cells. *Sci Immunol* **4**, doi:10.1126/sciimmunol.aau9039 (2019).

3 This work.

In addition, in the revised manuscript we have performed a rigorous comparison to the more recently proposed disulfide approach (Saini *et al.*, *Science Immunol.*, 2019). Consistently, we find that our method offers an improvement by a factor of 10 in terms of final pMHC yield, after the refolding, purification and exchange steps (Supplementary Table 1). This is due to the fact that the refolding of disulfide-linked, empty molecules is inherently more difficult, even in the presence of dipeptide in the refolding buffer according to the protocol provided in the recent *Science Immunology* paper. In contrast, the refolding of MHC complexes with goldilocks peptides, which is the basis of our approach, is highly efficient and can yield several milligrams of purified protein.

While these improvements over existing methods might seem incremental, considering that the production of typical sized pMHC libraries involve 100s of different peptide exchange reactions this boost in efficiency is highly relevant for enabling robust applications to probe polyclonal TCR repertoires, which is the main focus of our work. These points have been included in the revised manuscript (p 6, pp 2 & p 7).

In addition, it is unclear whether the metrics they show include the spin column dialysis that they describe needing to fully remove TAPBPR.

We have repeated the staining experiments using new tetramers prepared from 1:1 stoichiometric TAPBPR:MHC complex followed by TAPBPR removal after the peptide exchange reaction by spin column dialysis with 1000x buffer. Complete TAPBPR removal was confirmed by SDS-PAGE gel, as shown in Supplementary Figures 6&7. In addition, we have developed a new, simplified protocol using catalytic amounts (1%) of TAPBPR in an overnight peptide exchange reaction which can be removed in a single wash with 10x buffer (Supplementary Fig. 4). Both protocols are outlined in detail in the revised Methods section (p 27-28).

The new staining results, shown for three different cell lines in the revised Fig. 3, demonstrate that TAPBPR-exchanged tetramers produced using either the "stoichiometric" or "catalytic" approaches are nearly identical in terms of staining efficiency to the "industry standard" refolded tetramers, with the added benefit that they can be produced for 100s of peptides at high-throughput.

Supplementary Figure 4. Native gel of HLA-A*2:01 pMHC following peptide exchange reactions catalyzed by TAPBPR for peptides of different charges.

REVIEWERS' COMMENTS:

Reviewer #1 (Remarks to the Author):

The authors have nicely addressed my technical concerns. I suppose time will only tell whether the immunology community determines whether this TAPBPR mediated exchange or the disulfide-stabilized MHC-I molecules is the easier approach. Nevertheless, rapid and reliable MHC-I tetramer libraries is an advancement for the community.

Reviewer #2 (Remarks to the Author):

The authors have done an excellent job of addressing the comments, including a demonstration of what in their work is novel. Further, the authors should be commended for pointing out advances in the field since their initial submission that affect interpretation of their results. In general, the manuscript is well-written and interesting. The manuscript has improved in response to critiques.

REVIEWERS' COMMENTS:

Reviewer #1 (Remarks to the Author):

The authors have nicely addressed my technical concerns. I suppose time will only tell whether the immunology community determines whether this TAPBPR mediated exchange or the disulfide-stabilized MHC-I molecules is the easier approach. Nevertheless, rapid and reliable MHC-I tetramer libraries is an advancement for the community.

We wish to thank the reviewer for their positive appraisal of our work and its significance for the broader Immunology community.

Reviewer #2 (Remarks to the Author):

The authors have done an excellent job of addressing the comments, including a demonstration of what in their work is novel. Further, the authors should be commended for pointing out advances in the field since their initial submission that affect interpretation of their results. In general, the manuscript is well-written and interesting. The manuscript has improved in response to critiques.

We wish to thank the reviewer for their comments, and truly appreciate their constructive criticism.